# Evaluation of Foot and Claw Lesions and Claw Horn Growth in Piglets from Birth to End of Nursery

**DOI:** 10.3390/ani13223477

**Published:** 2023-11-10

**Authors:** Maren Bernau, Hannah Meckel, Theresa Dölle, Armin Manfred Scholz

**Affiliations:** 1Livestock Center Oberschleissheim, Veterinary Faculty, Ludwig-Maximilians-University Munich, St. Hubertusstrasse 12, 85764 Oberschleissheim, Germany; meckel@tierarztpraxis-hohenlohe.de (H.M.); theresa-doelle@web.de (T.D.); a.s@lmu.de (A.M.S.); 2Hochschule für Wirtschaft und Umwelt Nürtingen-Geislingen, Fakultät Agrarwirtschaft, Volkswirtschaft und Management, Nuertingen-Geislingen University, Neckarsteige 6-10, 72622 Nuertingen, Germany; 3Tierärzte Hohenlohe Dres. Wesselmann & Zankl Partnerschaft, Bölgentalerstraße 5, 74599 Wallhausen, Germany

**Keywords:** swine, foot lesion, scoring, claw horn length

## Abstract

**Simple Summary:**

Foot lesions of piglets are reported in several articles and might be linked to floor conditions or sow performance. Therefore, the present study evaluated claw horn growth and evaluated foot lesions and bruising from the day of birth until the end of the suckling period in three different genetic lines. Differences among genetic lines were detected.

**Abstract:**

The objective of this study is to evaluate foot lesions and claw horn growth of piglets from the day of birth until the end of the suckling period and describe the prevalence and extent in three different genetic lines. Therefore, bruising, dorsal horn lesions, claw horn growth, and weight gain were evaluated five times during growth, starting with the day of birth (day 0 of life) and ending with the end of nursery (day 68 ± 2 of life). Totally, 74 piglets of three genetic lines (German Landrace × Large White; Piétrain × Piétrain-Duroc; pure German Landrace) were examined. Bruising and dorsal horn lesions reached maximum levels at day 7 (±1) of life (with up to 91% of piglets having bruising marks and up to 94.1% of piglets having dorsal horn lesions). Differences among genetic lines were detected, with German Landrace × Large White crossbreds showing the highest percentage of bruising, but Piétrain × Piétrain-Duroc crossbreds showed the highest score for dorsal horn lesions at day 0. Until weaning (day 28 ± 1), front feet were more affected by bruising than hind feet (70.3% of the front feet and 64% of the hind feet showed bruising), but at the end of nursery (day 68 ± 2), hind feet showed a higher percentage of affected feet than front feet (65.5% vs. 41.3%). Several factors affect bruising scores in piglets, including body weight, age at examination, litter size, sex, parity, breed, and claw horn length. Additionally, significant differences for claw horn length were detected among the genetic lines from birth to end of nursery.

## 1. Introduction

Foot lesions in piglets have been discussed for several years and remain a problem at present [1]. Claw horn lesions could be worsened by an infection resulting in even more severe welfare and economic problems. One of the most commonly observed lesions in pre-weaning piglets is sole bruising, with prevalence between 50 and 100% [2,3,4,5,6]. Sole bruising is dominantly described as dark red pigmentation on the palmar/plantar surface of the foot [2]. It is often discussed as a result of flooring malfunctions [7,8,9,10,11], mainly discussed in growing pigs. Additionally, claws of newborn piglets are made of soft horn tissue, which may lead to a higher vulnerability [2]. Several aspects can be discussed, but foot lesions and bruising occur already in newborn piglets in different flooring types at the time of birth combined with other inflammatory and necrosis signs at the tail, ears, nipples, and navel [12,13,14].

Regardless of the cause, claw and limb lesions result in lameness, pain [8], and thus less active piglets [2] that achieve lower daily gains [15], and these piglets may have a higher risk of being crushed because several factors affect piglet viability [16]. Therefore, it is of major interest in today’s piglet production when it comes to animal welfare.

Additionally, little is known about claw horn growth in piglets. Van Amstel and Doherty (2010) [17] found a discrepancy between inner and outer claw size on the hind feet of three finishing pigs (113–150 kg body weight). Newton et al. (1980) [18] examined a floor effect on claw horn length in growing to finishing pigs. Also, in growing to finishing pigs, Johnston and Penny (1989) [19] examined diminished horn growth with age. Van Riet et al. (2016) [20] examined claw horns from the fourth to the ninth week under a zinc dietary treatment. They found differences between the lateral and medial claw digit regarding horn quality [20]. As claw length is associated with growth, there is a link to performance, and with this, there might be a link to foot lesions, as discussed with sow performance [21,22]. Additionally, Seufert et al. (2022) [1] demonstrated a link between strongly asymmetric claws and the risk of sole ulcers in pigs.

Therefore, the present study evaluates claw and foot lesions from the day of birth until the end of nursery in three different genetic lines divided in eight litters. Additionally, weight gain and claw horn growth are evaluated in order to check the association with claw lesions.

## 2. Materials and Methods

### 2.1. Animals and Housing Conditions

The data presented in this study were from 74 piglets that were to be examined 5 times from birth (day 0) to the end of nursery (day 68). However, only 58 pigs survived until the end of nursery on day 68. Seven pigs died or had to be euthanized between birth and day 3. Another 3 pigs died between day 7 and day 28, while 6 pigs did not survive the growth phase between day 28 and day 68 of the study.

Examinations took place from March to July 2017 in a conventional pig farm in Bavaria, Germany. The herd consisted of maximum 140 sows of different genetic lines (with an average age of 3.8 years) and a performance of 23 weaned piglets per sow and year. From each farrowing group, three (or two (for GL-LW)) sows were selected randomly that had the same genetics, regardless of the parity number, resulting in purebred or crossbred piglets of 3 different genetic lines, as outlined in Table 1.

The farrowing unit was built of 10 farrowing units in one compartment. One farrowing group consisted of a maximum of 20 sows. Totally, seven groups of maximum 20 sows per group rotated in a three-week production rhythm through the farrowing, artificial insemination, and gestation units. The replacement rate for multiparous sows was 30% per year. Sows were kept in farrowing pens (2.5 m × 1.8 m) with farrowing crates (0.55–0.70 m wide, 1.50–2.0 m long, 1.15 m high) from one week before expected farrowing until weaning (lactation period 28 days). The floor was built of two polypropylene floor types: “Pigfloor A” (with 9.8 mm slats and 12 mm slots, perforation 35%) and “Pigfloor C” (with strongly reduced perforation of <3.5%). Pigfloor “C” covered 15% of the farrowing unit (without accounting for the piglet “nest”). In addition, piglets had free access to another polypropylene “warm piglet nest” area covering 0.62 m^2^ of the 4.5 m^2^ farrowing unit without any perforation. This piglet “nest” was heated by warm water through an underfloor hose system and was optionally equipped with a heating lamp.

Sows were vaccinated against the porcine parvo virus, *Erysipelothrix rhusiopathiae* (both before mating), and the influenza virus as well as *Clostridia* spp. (before farrowing). Piglets were vaccinated against *Mycoplasma hyopneumoniae* and Porcine circovirus type 2 (21st day of life). Piglets were tail docked by cutting and male piglets were castrated at day 4 ± 1 using pain relief. Sows were fed 14.7 MJ ME/kg dry matter (DM) after entering the farrowing unit. Piglets were fed with milk exchanger (20 MJ ME/kg DM) alongside the mother´s milk from day 4 of life and they received supplementary feed (15.1 MJ ME/kg DM) from day 8 on. After weaning piglets received a diet containing 15.4 MJ ME/kg DM. Weaning was performed on day 28 of life. Sows and piglets were moved to different compartments. The farrowing compartment was then cleaned, disinfected, and left empty for one week.

After weaning, the piglets were moved to the nursery deck. Each deck offered an area of 15.83 m^2^ and used “Pigfloor F” (with 10 mm slats and 10 mm slots, perforation 38%) and “Pigfloor FC” (with 10 mm slats, perforation < 10%), which was used in a ratio of 64 to 36%. All floor types were built by “Mannebeck Landtechnik GmbH” (Quendorf, Germany), and each housing system is in accordance with local and national guidelines [23,24].

### 2.2. Data Sampling

All piglets of a selected litter were examined five times regarding claw health (see ‘claw check’ at day 0, 3 (±1), 7 (±1), 28 (±1), and 68 (±2); Figure 1) with focus on bruising and horn lesions. First claw check (day 0) was performed within two hours after birth. The left front and the left hind claw, including inner and outer claw, were evaluated, and the location of alteration—palmar and dorsal claw site—was recorded for each claw.

#### 2.2.1. Foot and Claw Lesions

Dorsal horn lesions and bruising were assessed using a scoring from 0 (no alteration) to 5 (severe alterations) separately for front and hind feet and for inner and outer claws, resulting in four scores for dorsal horn lesions and eight scores for bruising. The scoring was performed by one trained person. Bruising was defined as dark red pigmentation on the palmar or dorsal surface of the claw (Figure 2a–c, sole bruising). The total size of alteration was scored in dorsal and palmar location with a score from 0 to 5 (Table 2). Dorsal horn lesions were defined as alterations in the claw horn surface, focusing on grooves (Figure 2d), horn fissures (Figure 2e), and injuries/bleeding in the coronary band (Figure 2f), excluding cracks, scored from 0 to 5. Table 2 summarizes the different scores.

#### 2.2.2. Weight and Weight Gain

At day 0 (birthday), before first claw check, the animals were ear tagged and weighed, within two hours after birth. The weight of the individual animal was recorded on all subsequent days of the study (day 3 (±1), 7 (±1), 28 (±1), and 68 (±2)) to calculate the daily gain (g) as a performance trait. Average daily gains were calculated by dividing the difference between body weight records at the days of the data recording with the age difference.

#### 2.2.3. Claw Horn Length

Additionally, claw horn length (cm) was measured five times until the end of nursery (Figure 1: ‘horn length’ at day 0, 3, 7, 28, and 68) from the coronary band to the apical end of the wall by using a caliper (demonstrated in Figure 2g as blue lines). The length of the claw horn was recorded on the left front and the left hind claw, including inner and outer claw, respectively.

### 2.3. Statistical Analysis

The data were analyzed using SAS 9.4 software (SAS Institute, Cary, NC, USA).

Foot and claw lesions: To evaluate the claw check data, assessed as scoring values between 0 and 5, the FREQ(UENCY) procedure was used. A Fisher’s exact test was performed by means of a Monte Carlo calculation for the association between measurement day and score frequency (for claw lesions and bruising), and for the association between genetic line and score frequency (for claw lesions and bruising) by measurement day.

Weight and claw horn length measurements: Additionally, a repeated measure mixed model variance analysis (PROC MIXED) using a restricted maximum likelihood estimation (REML) with the fixed-effects genetic line, sex, measurement number, interaction of measurement number × genetic line, and animal number as repeated effect was performed to evaluate claw horn length and body weight data at the time of measurement.

The growth data (average daily gain) were analyzed by a modified mixed model separately for every growth period (time between two measurement points) with the fixed effects genetic line and sex.

In all cases, the parity number was used in a first version of the mixed model analysis as random affect for the weight, growth, and claw data but did not show a significant effect within the models. Therefore, parity number was excluded from the final model. Birth weight was not used as covariate in the mixed model analysis.

The significance level was set to *p* ≤ 0.05 for all calculations.

To calculate the interfering effects regarding the bruising scores, a logistic stepwise regression was used with the fixed-effects breed, sex, age at examination (measurement number), and parity (first vs. multiparous). As covariates, claw horn length, body weight, and litter size were used. The significance level was set to *p* ≤ 0.05 for entering and remaining in the regression analysis. After the stepwise regression, odds ratios were calculated and visualized as plots for the resulting logistic regression models (SAS/STAT^®^ 13.1 User’s Guide, The LOGISTIC Procedure, SAS Institute, Cary, NC, USA).

## 3. Results

### 3.1. Foot and Claw Lesions

#### 3.1.1. Frequency Analysis

Figure 2 shows photographs of detected lesions (a–c = sole bruising; d–f = dorsal horn lesions). For bruising, significant associations exist between the proportion of score values (0–5) and examination days depending on the claw location. During the examination period, the percentage of animals with score 0 decreases until day 7 (Figure 3). On day 0, bruising is found in a maximum of 25% of the animals, with a focus on the palmar surface of especially the front claws (score 2 = 13.4% of the animals for inner claws and 10.5% of the animals for outer claws). On day 3, the percentage of affected feet (scores > 1) increases (palmar front feet = maximum of 80.8% of the animals; palmar hind feet = maximum of 83.8% of the animals), reaching the maximum score at day 7. On day 7, the highest scores are detected (a maximum of 55.2% of the animals showing a score ≥ 3), with up to 91% of the piglets showing bruising in the front and hind claws (palmar surface). On day 28, the percentage of affected palmar claw surfaces decreases in front claws to 70.3% (inner claw) and in hind claws to 64%. The percentage of affected dorsal surfaces increases at that time. At the end of nursery (day 68), in the front claws, bruising is still detected in 41.3% of the animals, whereas in the outer hind claw, 65.5% of the animals show bruising (palmar surface).

Besides the general findings, partially significant associations among the genetic lines and score frequencies by evaluation day are detected (Appendix A). Up to 45% of GL and Pi-PiDu piglets show bruising, with a maximum score of 2 at the palmar surface of both claws (with a higher percentage on front rather than on hind claws) at day 0. GL-LW shows the least claw alterations on day 0. On day 7 of life, up to 95% of the piglets of GL and Pi-PiDu and up to approximately 88% of the GL-LW piglets show bruising at the palmar surface of the front and hind claws (inner and outer claw). GL and GL-LW have a maximum score of 5, whereas Pi-PiDu received a maximum score of 4 (with up to 29.2%). GL shows also bruising at the dorsal surface of both feet.

For dorsal horn lesions, significant associations exist between the proportion of score values (0–5) and examination days depending on the claw location (inner versus outer claw; Figure 4). Figure 2 demonstrates photographs of the different type of lesions, which were summarized as dorsal horn lesions: (d) = grooves in the horn surface, positioned mainly laterally and medially at the horn surface; (e) = horizontal fissures at the surface of the horn capsule; (f) = coronary band bleeding/injuries. Until weaning, the outer claw seems to be more often affected than the inner claw (outer claw with a lower score of 0). On day 0, 46.3% (inner front feet) to 67.2% (outer front feet) of the piglets show dorsal horn lesions (mainly grooves). The maximum of lesions is found on day 7, with 71.6% (inner hind claw) to 94.1% (outer front claw) of the piglets showing some kind of dorsal horn lesion. This percentage decreases afterwards from weaning (day 28 ± 1 = 45.3–57.8%) towards the end of nursery (day 68 ± 1 = 18.9–27.6%; Figure 4).

Additionally, significant associations between score frequencies and genetic lines appear for dorsal horn lesions (Appendix A). Most lesions exist on day 7 of life, whereas for Pi-PiDu, the highest scores emerge on day 28 (weaning). Pi-PiDu also has the highest percentage of affected claws on day 0 (more than 95%), whereas GL has the lowest score (maximum of 30%).

#### 3.1.2. Results of Logistic Regression for Bruising

As shown in Figure 5, several factors affecting bruising scores in piglets were found. On the palmar surface, for both front and hind claws, body weight, age, and litter size can be identified as effects on the bruising score. The lighter the animals, the younger the age at examination, and the smaller the litter size, the worse the score, i.e., higher scores are obtained. In addition, there is a sex effect (outer claw), with worse scores for females, and a breed effect (inner claw), with worse scores for GL and GL-LW compared to Pi-PiDu.

Additionally, for the dorsal surface, claw length (front and hind claw), body weight (inner front claw), parity (outer front claw), breed (inner and outer hind claw), litter size (outer hind claw), and age at examination (inner hind claw) were identified as affecting variables with a similar outcome as for the palmar surface (see Figure 5).

The claw length entered the logistic regression equations for all hind leg claws. A 1 cm shorter claw was always associated with a decrease in the odds ratio of at least 22% of having no bruising, i.e., a bruising score of 0 (see Figure 5; hind feet inner claw dorsal surface with a confidence interval for the odds ratio of 0.018 to 0.773). Shorter claws on the hind feet posed an even greater problem for the inner claws than for the outer claws.

### 3.2. Weight and Weight Gain

The analysis of the weight data shows significant differences among the genetic lines only at the end of nursery (day 68; Table 3). GL-LW shows besides the smallest final weight also the smallest average birth weight with 1.1 kg. These piglets originate from the largest litters (Table 1). In addition, the mean body weight of females is significantly (*p* ≤ 0.05) higher than that of males (7.62 ± 0.13 kg vs. 7.15 ± 0.13 kg).

The analysis of the daily gain (g) shows no significant differences among the genetic lines, but it does show a gender effect for the time period day 7–28 (Table 4). GL-LW shows the highest daily gain until day 7. The Pi-PiDu show the highest daily gain from day 0 to 68 (not significant).

### 3.3. Claw Horn Length

Claw horn length differs between breeds (Table 5). At birth (day 0), outer claws of GL-LW are shorter than for Pi-PiDu, though the claw horn length of the front inner and outer claws reaches the greatest value for GL-LW at the end of nursery (day 68). Therefore, the largest front horn growth occurs in GL-LW combined with the least front horn growth in GL between days 3 and 68. Significant differences were observed from day 0 until the end of nursery (day 68) and at days 0 and 3, mainly on the hind legs and especially at the outer claw (Table 5). Male and female piglets do not differ significantly regarding claw horn length.

The highest claw horn growth was detected between days 7 and 28 (Figure 6). Additionally, hind claws gain more length than front claws, especially from day 28 on (Table 5, Figure 3).

## 4. Discussion

The data presented in this study origin from 74 piglets, which were planned to be examined five times from birth (day 0) until the end of nursery (day 68). However, only 58 pigs survived until the end of nursery on day 68. The greatest loss of pigs (7 from 17) occurred in a GL-LW litter between day 0 and day 68. All examinations, including claw health scoring and horn length measurements, were performed by one trained individual in order to reduce inter-observer variation. Although it was only a small number of animals, significant differences were found, confirming previous studies on piglet feet.

### 4.1. Foot and Claw Lesion

Bruising: Bruising was defined as dark red pigmentation on the surface of the foot, as described by Mouttotou et al. (1999) [3]. This red pigmentation was seen in up to 25% of the animals on their birthday (day 0), with a higher percentage in the front legs (Figure 3). These findings cannot only be traced back to the flooring, as this is of minor interest if scoring was performed within 2 h after birth. The results of the present study confirm several previous studies [2,3,4,5,6,25], with incidence of bruising increasing with age and reaching a maximum level on day 7. In most of the studies, these lesions were seen from day 1 to day 28 [3,6]. In the present study, the maximum percentage of affected feet was recorded on day 7 (with up to 91% of piglets having bruising marks). It can be proposed that mechanical causes like flooring types might not be the only reason for this alteration, as it is more likely that internal factors [12,13,14] or the performance of the sows [21,22] might be interfering factors as well. This can be confirmed by the present study (see odds ratios in Figure 5), where several factors have an effect on bruising score.

Mouttotou et al. (1999) [3] did not find differences between front and hind feet until day 28. In the present study, on day 28, 70.3% of the front feet and 64% of the hind feet showed bruising, whereas on day 68, 41.3% of the front feet but up to 65.5% of the hind feet showed bruising, meaning a frequency shift of feet diagnosed with bruising from the front to the hind feet (Figure 3). Mouttotou and Green (1999) [2] examined a mean sole bruising duration of 13 days. In the present study, it seems that bruising occurs over a longer period than 13 days (from day 0 to day 68 in different severity). This can be confirmed by the present study using the odd ratios, where the age at examination affects bruising scores, which leads to younger piglets having higher scores than older piglets (Figure 5).

The differences among genetic lines/litters in the present study cannot be traced back to different flooring conditions, because all sows and piglets were kept under the same conditions. Interactions might exist between litter size, birth weight, weight development, health status of the sow, milk quality, and/or environmental conditions. This can also be confirmed by the present study (Figure 5), where influencing factors for bruising are litter size, weight, age at examination, and parity of the sow. Therefore, further studies are necessary, especially as bruising might potentially cause pain and discomfort and is therefore an animal welfare risk [1,2,5,25].

Dorsal horn lesions: The term ‘dorsal horn lesions’ covers grooves, horizontal fissures, and coronary band bleedings/injuries. The most lesions were detected on day 7, with up to 94.1% of piglets (Figure 4). A decrease in percentage of affected claws, however, was firstly recorded after day 0 until day 3. This can be explained with the presence of grooves, which were mainly detected on day 0. This might be traced back to the physiological loss of the horn cowl after birth. The maximum of detected lesions was found at day 7, which can be explained by active piglets fighting for milk. Therefore, the flooring type might be the reason for this increase in alterations, as this was mainly due to coronary band bleeding or injuries. Further studies are necessary with more animals and a separated presentation of each possible dorsal horn lesion with an exact definition. Additionally, weight and activity as well as weight and floor interactions should be kept in mind as effects. Future studies should evaluate whether weight and activity are connected and whether this might result in more trampling and fighting and, with this, more alterations.

After day 7, the percentage of affected claws decreased again (Figure 4). On all examination days, the outer claw was more often affected then the inner claw. This finding confirms previous studies [7,17]. Van Amstel and Doherty (2010) [17] found a discrepancy between inner and outer claw size on hind feet. The authors explained this finding by the fact that the outer claw of the hind leg has to bear more weight relative to the inner claw, and therefore has to bear more of the mechanical insult related to locomotion. This observation might result in increased claw horn growth and wear rates for hind feet, resulting in less mature keratinized horn cells finally predisposing a development of claw horn lesions. In the present study, differences between inner and outer claw were found at front and hind legs (Table 5, Figure 4). This can be confirmed with the findings of van Riet et al. (2016) [20], who found differences between lateral and medial claw digits regarding horn quality [20]. A difference in horn quality might increase claw horn lesions and abrasions. Additionally, Seufert et al. (2022) [1] demonstrated that the occurrence of asymmetric claws increase over the pigs’ lifetime and that this results in a higher risk for sole ulcers. Therefore, further studies are needed, focusing on claw horn quality, claw horn growth, and asymmetry of claws, as they may be factors for horn lesions and therefore pain and discomfort in animals.

Additionally, differences among the different genetic lines occurred in the present study (Appendix A; Figure 5). On day 0, Pi-PiDu showed the highest level of affected claws compared with the other two lines. GL showed the lowest percentage of affected claws on day 0. The maximum percentage of affected claws was recorded on day 7, and the highest score was achieved at day 28. These differences might be traced back to different horn strength or hardening during weight gain or claw horn growth. Therefore, more animals have to be examined, and additionally, the horn quality needs to be examined on a genetic level.

Furthermore, activity and lying time might be a result of pain and discomfort due to foot lesions [6,25] and should be examined in future studies as well. To our knowledge, there is no direct comparison of bruising scores or feet alterations with lameness, welfare, pain, or economic values. Jensen et al. (2012) [26] quantified the impact of lameness on welfare and economic impact by expert opinions, but a study involving clinical signs and behavioral data compared to pathomorphological examinations is still lacking.

### 4.2. Weight Measurement

No significant differences were detected for average daily gain between the genetic lines (Table 4), though body weight differed significantly at the end of nursery on day 68 (Table 3). GL-LW were the lightest piglets until day 3 but had the highest average daily gain (Table 4). GL-LW had an average litter size of 15.3 ± 2.0 piglets (born alive). Therefore, the cause of the lightest average body weight occurring during the first days of life could be backtracked to more piglets born. These piglets showed the highest average daily gain (not significant) and, in addition, the least percentage of bruising at first claw check (day 0, Appendix A). Logistic regression showed an effect of litter size, where piglets from larger litters tend to have increased odds of having lower bruising scores (Figure 5, e.g., front feet inner and outer claws each palmar, hind feet inner claws dorsal, as well as hind feet outer claws palmar). This might be linked to high sow performance, which has to be confirmed by larger animal numbers and by recording performance data in more detail.

The heaviest piglets during the suckling period were GL. These piglets had the highest bruising scores at day 0 and day 7 (Appendix A), which could be a hint towards bruising affecting weight gain, as discussed by Mouttotou and Green (1999) [2]. But, as shown in Figure 5, a higher body weight is associated with lower bruising scores. Therefore, the highest scoring results of GL piglets might be a result of a breed effect (demonstrated in Figure 5), as GL and GL-LW had decreased odds for lower bruising scores than Pi-PiDu.

Mouttotou and Green (1999) [2] described that sole bruising during the first days of life is associated with higher weight. This can be confirmed even with a small number of animals in the present study, as the genetic line with a higher weight (GL and Pi-PiDu) showed the highest score degree regarding bruising on day 7 (Appendix A). This has to be analyzed with larger animal numbers, as the results of the logistic regression demonstrated an effect of body weight, with higher body weights resulting in increased odds of lower bruising scores, especially at the front feet inner claws (Figure 5). But, on the other hand, Gillman et al. (2009) [25] stated that 14-week-old piglets are not heavy enough that weight can be a significant factor in lesion development.

Additionally, using the results of the stepwise logistic regression analysis (Figure 5), a breed effect was demonstrated, showing GL and GL-LW having decreased odds of lower bruising scores than Pi_PiDu at the hind feet inner claws and at the hind feet outer claws on the dorsal side, which might be the reason for the above-stated findings. But, as this study only has a limited number of animals, further studies are necessary.

Bruising was found at day 0 in the present study (up to 25%), with highly increasing levels of bruising alterations on day 7 (up to 91%) that decrease afterwards but are still present (up to 65.5% of the animals at the end of nursery (day 68)). According to the recent literature, e.g., [12,13,14], the observed alterations might be one part of the swine inflammation and necrosis syndrome (SINS). With this, maybe more piglets per litter are a link to a healthier sow (at mating time) and therefore result in less bruising as a result of higher sow performance.

It has to be kept in mind that these sole alterations and lesions (Figure 4)—seen during the suckling period with maximum levels at day 7—could potentially cause pain and discomfort and may decrease the general activity of the piglets and the associated suckling activity [2,9,25]. Although the present study included a relatively small number of animals, factors influencing bruising scores in piglets were detected (Figure 5), including body weight. Therefore, more studies with larger numbers of animals and a homogeneous genetic background are needed, combined with behavioral documentation, to detect more factors for more bruising or lower weight, or even average daily gain in different breeds. Additionally, other environmental factors have to be recorded as well, like the well-being of the sow, milk quality and quantity, stress triggers, and performance parameters of the sow, as it could be demonstrated with this small animal number that sow performance parameters may affect bruising scores in piglets (Figure 5). As claw health in sows is linked to (reproductive) performance [21,22], this should be examined as well, to evaluate the maternal influence (sow parity) on the growth performance of piglets.

### 4.3. Claw Horn Length and Growth

The present study found significant differences among the genetic lines until day 68 (±2) of life, with GL-LW having the shortest claw horn length (Table 5; at some days together with Pi-PiDu), but also having the lightest piglets (Table 3), whereas GL had the longest claw horn (Table 5, front and hind legs), combined with the heaviest piglets. At the end of nursery (day 68), significant differences among the genetic lines were still present, especially with the hind claws being longer than front claws (Table 5; Figure 6). At that day, differences between inner and outer claws were observed for GL, but only on hind feet. This finding confirms the study of van Amstel and Doherty (2010) [17], who found a discrepancy between inner and outer claw size on hind feet (three pigs, 113–150 kg body weight). Additionally, the present study can confirm the findings of Johnston and Penny (1989) [19] regarding diminished horn growth with age—as demonstrated in Table 5 and Figure 6, with horn growth being higher during suckling and lower after weaning.

Additionally, using the results of the stepwise logistic regression (Figure 5), claw horn length was detected as one factor influencing the odds of lower bruising scores. GL-LW had the shortest claw horn length (Table 5), which is associated with decreased odds for lower (better) bruising scores. Therefore, further studies are needed to evaluate claw horn growth in different breeds and compare these results to bruising scores and piglet performance as well as sow fitness to be able to identify the most affecting parameter. As shown by Seufert et al. (2022) [1], the asymmetry of the claws should be taken into account as well, as this seems to be one risk factor for sole ulcers. The results of Seufert et al. (2022) [1] can be partly confirmed with the present study, as shorter claws on the hind feet posed an even greater problem for the inner claws than for the outer claws.

## 5. Conclusions

To sum up, weight development, claw horn growth, and the odds of lower bruising scores seem to be linked. Differences in growth and length between front and hind claws were detected. Also, a link between weight gain and foot lesions was detected, although only a small number of animals was evaluated. Bruising was detected two hours after birth; therefore, floor conditions cannot be solely responsible for this alteration. The stepwise logistic regression analysis showed that several factors influence bruising scores in piglets, including piglets’ weight, age, sex, claw horn growth, sow parameters (parity, litter size), and breed or crossbreed combination. Further studies with larger numbers of animals and a focus on sow performance, sow health, sow parity, and claw horn growth and quality in different genetic lines are necessary.

## Figures and Tables

**Figure 1 animals-13-03477-f001:**
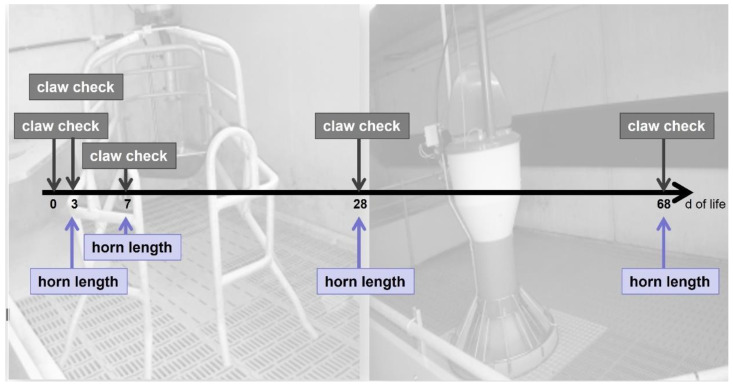
Examination schedule from birth (day 0) to end of nursery (day 68 ± 2), including pictures of the floor conditions. ‘Claw check’ included examination of bruising and dorsal horn lesions. ’Horn length’ was measured by a caliper.

**Figure 2 animals-13-03477-f002:**
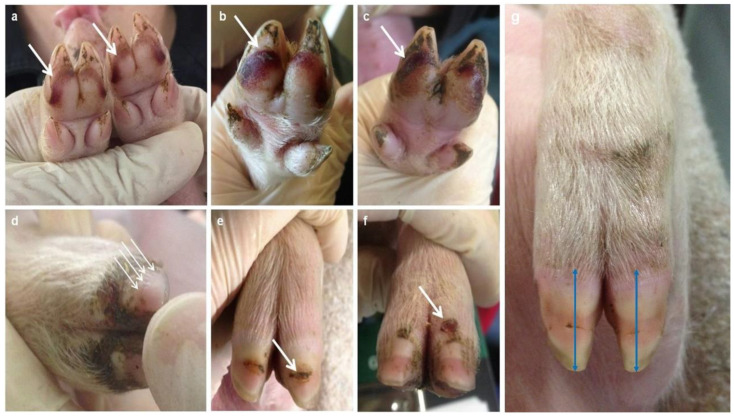
Photographs of the detected foot alterations, with alterations highlighted with a white arrow: (**a**–**c**) = sole bruising ((**a**) = piglet at day 0); (**d**–**f**) = dorsal horn lesions ((**d**) = grooves, (**e**) = horizontal fissures, (**f**) = coronary band bleeding/injuries). Photograph (**g**) demonstrates the measurement of claw horn length on the dorsal surface of the inner and the outer claw (demonstrated as blue arrows).

**Figure 3 animals-13-03477-f003:**
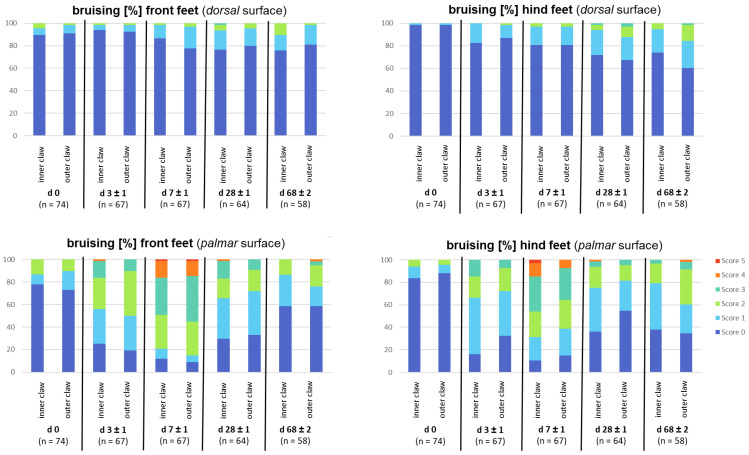
Mean proportion (%) of claw score ‘0’ for bruising during growth between birth and end of nursery (day 68), displayed for front and hind feet, dorsal and palmar surface, as well as inner and outer claw.

**Figure 4 animals-13-03477-f004:**
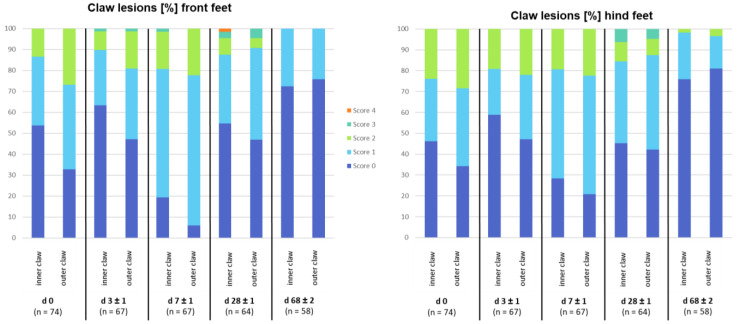
Claw lesion score results, differentiated according to days of life and locations. Fisher’s exact test (using Monte Carlo estimation) was significant with *p* < 0.0001 for all four tests or dorsal claw lesion sites (testing the association between claw lesion score frequency with scores from 0 to 5 and measurement day 0, 3, 7, 28, 68).

**Figure 5 animals-13-03477-f005:**
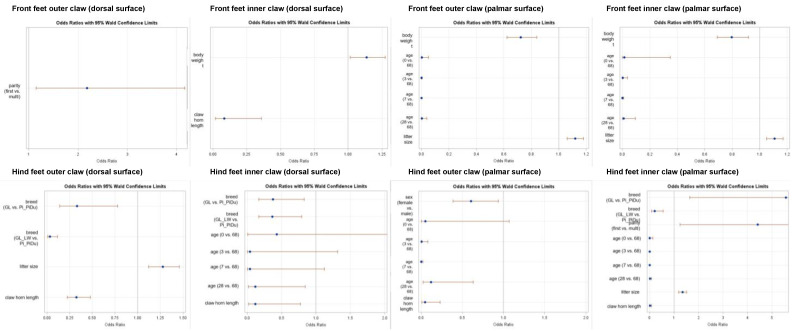
Results of stepwise logistic regression analyses, differentiated for front and hind claws, outer and inner claws, and dorsal and palmar surfaces to demonstrate effects of variables affecting the odds of having lower bruising scores.

**Figure 6 animals-13-03477-f006:**
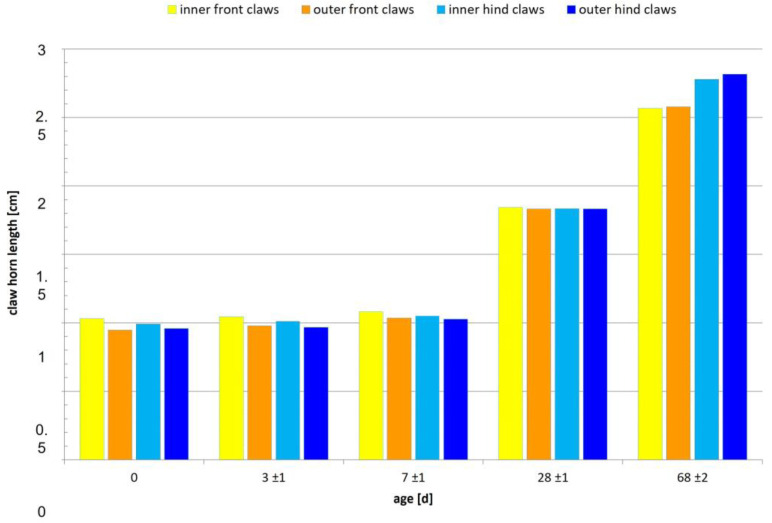
Claw horn length [cm] displayed from day 0 to end of nursery (day 68 ± 2), differentiated into front and hind legs and in inner and outer claws. Vertical bars are indicators of the estimation errors.

**Table 1 animals-13-03477-t001:** Description of experimental piglets divided according to their genetic origin.

“Genetic Line”	Number of Piglets Examined on Day 0|3|7|28|68	Number of Litters Included	Number of Piglets (Born Alive) per Litter	Piglet Sex (Male:Female) on Day 0
German Landrace × Large White (GL-LW)	30|26|26|23|18	2	15.3 ± 2.0	12:18
German Landrace (GL)	20|19|19|19|19	3	7.9 ± 2.5	12:8
Piétrain × Piétrain-Duroc (Pi-PiDu)	24|22|22|22|21	3	8.6 ± 2.3	11:13
**SUM/Total**	**74|67|67|64|58**	**8**	**11.1 ± 4.1**	**35:39**

**Table 2 animals-13-03477-t002:** Scoring bruising (sole bruising and on dorsal surface) and dorsal horn lesions from 0 to 5.

Lesion	Score 0	Score 1	Score 2	Score 3	Score 4	Score 5
**bruising**	absence of bruising	mild and small alterations	≤25% of the claw covered	≤50% of the claw covered	>50% but ≤75% of the claw covered or multiple areas	>75% of the claw covered
**dorsal horn lesion**	no lesion	minimal	≤0.5 cm	>0.5 cm but ≤1 cm	>1 cm	>2 cm

**Table 3 animals-13-03477-t003:** Weight development (kg) (least-squares means ± standard errors of estimation), differentiated for the different “genetic lines” and days of life. Different superscripts demonstrate significant differences (*p* < 0.05) among LSM within a row.

Day of Life		Weight (kg) LSM ± SEE	
	GL	GL-LW	Pi-PiDu
day 0	1.66 ± 0.37	1.10 ± 0.30	1.50 ± 0.34
day 3 (±1)	1.98 ± 0.38	1.52 ± 0.34	1.79 ± 0.35
day 7 (±1)	3.25 ± 0.38	2.56 ± 0.34	2.66 ± 0.33
day 28 (±1)	8.83 ± 0.38	8.15 ± 0.34	8.16 ± 0.35
day 68 (±2)	22.69 ± 0.38 ^ab^	21.73 ± 0.39 ^a^	23.23 ± 0.36 ^b^

GL = German Landrace; GL-LW = crossbred of German Landrace boar and Large White sow; Pi-PiDu = crossbred of Piétrain boar and Píétrain-Duroc crossbred sow.

**Table 4 animals-13-03477-t004:** Daily gain (g) (least-squares means ± standard errors of estimation), differentiated for the different “genetic lines” and days of life. Different superscripts demonstrate significant differences (*p* < 0.05) among LSM within a row.

	Daily Gain (g) LSM ± SEE				
Time Period	GL	GL-LW	Pi-PiDu	Male	Female
day 0–3	139.9 ± 11.5	161.9 ± 9.9	144.9 ± 10.7	146.0 ± 9.3	151.8 ± 8.3
day 3–7	236.8 ± 13.6	245.2 ± 12.2	219.6 ± 12.6	235.5 ± 11.0	232.2 ± 10.0
day 7–28	268.2 ± 12.8	261.3 ± 11.7	259.7 ± 11.9	247.0 ± 10.0 ^a^	279.1 ± 9.8 ^b^
day 28–68	348.8 ± 18.1	330.9 ± 19.5	366.5 ± 17.2	335.6 ± 16.0	361.8 ± 14.1
day 0–68	308.5 ± 11.2	297.2 ± 12.1	319.5 ± 10.6	294.9 ± 9.9	321.9 ± 8.7

GL = German Landrace; GL-LW = crossbred of German Landrace boar and Large White sow; Pi-PiDu = crossbred of Piétrain boar and Píétrain-Duroc crossbred sow.

**Table 5 animals-13-03477-t005:** Claw horn length (cm) (least-squares means ± standard error of estimation), differentiated for the different genetic lines, days of life, and locations. Different superscripts demonstrate significant differences (*p* < 0.05) among LSM within a row.

Claw Horn Length (cm) (LSM ± SEE)					
Day	Location		GL	GL-LW	Pi-PiDu
0	front	inner	1.07 ± 0.08	0.99 ± 0.03	1.04 ± 0.04
		outer	0.94 ± 0.07 ^ab^	0.88 ± 0.03 ^a^	1.01 ± 0.03 ^b^
	hind	inner	0.98 ± 0.07	0.94 ± 0.03	1.01 ± 0.03
		outer	0.92 ± 0.06 ^ab^	0.90 ± 0.03 ^a^	0.98 ± 0.03 ^b^
3	front	inner	1.11 ± 0.04	1.01 ± 0.03	1.02 ± 0.04
		outer	1.02 ± 0.04 ^a^	0.90 ± 0.03 ^b^	1.02 ± 0.03 ^a^
	hind	inner	1.06 ± 0.03 ^a^	0.96 ± 0.03 ^b^	1.02 ± 0.03 ^ab^
		outer	1.00 ± 0.03 ^a^	0.88 ± 0.03 ^b^	1.02 ± 0.03 ^a^
7	front	inner	1.16 ± 0.04 ^a^	1.04 ± 0.04 ^b^	1.05 ± 0.04 ^b^
		outer	1.12 ± 0.04 ^a^	0.96 ± 0.03 ^b^	1.04 ± 0.03 ^ab^
	hind	inner	1.14 ± 0.03 ^a^	1.03 ± 0.03 ^b^	0.99 ± 0.02 ^b^
		outer	1.10 ± 0.03 ^a^	1.00 ± 0.03 ^b^	0.99 ± 0.03 ^b^
28	front	inner	1.89 ± 0.04	1.79 ± 0.04	1.85 ± 0.04
		outer	1.87 ± 0.04 ^a^	1.76 ± 0.04 ^b^	1.87 ± 0.04 ^a^
	hind	inner	1.88 ± 0.03 ^a^	1.75 ± 0.03 ^b^	1.86 ± 0.03 ^a^
		outer	1.88 ± 0.03 ^a^	1.74 ± 0.03 ^b^	1.87 ± 0.03 ^a^
68	front	inner	2.53 ± 0.04 ^a^	2.65 ± 0.04 ^b^	2.52 ± 0.04 ^a^
		outer	2.51 ± 0.04 ^a^	2.63 ± 0.04 ^b^	2.61 ± 0.04 ^ab^
	hind	inner	2.81 ± 0.03	2.76 ± 0.04	2.77 ± 0.03
		outer	2.88 ± 0.03 ^a^	2.77 ± 0.03 ^b^	2.80 ± 0.03 ^ab^

## Data Availability

Data is contained within the article.

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
