# Peer review of "Evaluation of Foot and Claw Lesions and Claw Horn Growth in Piglets from Birth to End of Nursery"

_animals, 2023, doi:10.3390/ani13223477_

Round 1

Reviewer 1 Report

Comments and Suggestions for Authors

In general

Thank you very much for letting me review the paper of this small but very well-organized study of 74 pigs of three different pig breeds in one (?) herd.

An overall comment would be, that the real interesting part of this paper is the ‘foot and claw lesion’- part. My suggestion for the next review would be to build the paper around this part.

The order of presenting your findings could be 1. Foot and claw lesions, 2. Weight and Weight gain, 3. Claw Length (I suggest to make this chapter short).

Use this order (1.,2.,3.) both in your M&M, Results and Discussion chapters.

A risk factor assessment of weight and breed and claw length on the risk of foot and claw lesions could be a fourth bullet pint.

Specific comments

 Title: Consider a briefer title which could also be adapted to your aim (L 60) i.e.  "Evaluation of foot and claw lesions and claw horn growth in piglets from birth to end of nursery" – this is just a suggested example.

 L. 12:                Piglet’s means “piglet is” please rephase i.e. Feet lesions of piglets

L. 12 + L. 16:    feet or foot? Singular or plural: Please be consistent throughout title and manuscript.

L. 20-30:           Please include result numbers in the abstract

L. 37-38:           most common lesion in total or among foot lesions?

L. 39:                “on the surface of the foot” Please use anatomical terms i.e dorsal/lateral, integument, horn,

L. 60-62:           GOOD! THIS is a central, very important part of your paper! You are looking at claw lesions – but also foot lesions. Include ‘foot lesions’ in your aim.

L. 63:                Materials and Methods:

Please mention country location of the herd and basic herd characteristics: size, herd health status, age of piglets at weaning, number of farrowing and nursery units, cleaning procedures and other subject which could impact the results.

                          How were the piglets/litters selected? By random or convenience?

                          When was the samplings performed (month/year)

Make an additional chapter where you present your outcome/response variables: Foot and claw lesions, Weight gain and Claw Length and how you measure them. Figure 2 and Table 2 would be part of this chapter

L. 72-73:           I don’t understand this sentence: Are 15 % of farrowing pens decorated with the type C floor?

L. 79:                Bacterial species names should be written in italics i.e. Mycoplasma hyopneumoniae.

L. 89:                I do not understand “phenotyping” of piglets. You could write “Data sampling” instead?

L. 92:                weighted -> weighed (without t)

Table 2:            Use exactly the same terms as in Figure 2: i.e. write “Sole bruising” in the column with ‘Lesions”

                          Leave out “Alterations” under “Score 1”

L.117:               Do you by “the end of the wall” mean the apical part of the claw?

                          In your results you mention measures of both inner and outer horn length

You could supplement your figure 2 with a drawing of how you measure.

Table 2+3:        Minimize the use of horizontal lines in tables. Please see examples from other papers.

L. 120:              Statistical analyses:

Please mention how you calculate average daily weight gain.

                          Please elaborate which statistics you are using for each outcome/response variable:  (weight, average daily gain, claw length, claw lesion).

                          In your models for weight and weight gain: do you take birth weight into account?  

L. 178:              Part 3.3 This part is really interesting and should be the central and most important  part in this paper. I suggest you make “Claw (and feet?) lesions” you chapter 3.1, and minimize emphasis on weight and claw horn length.

(Why is claw horn length interesting?  Perhaps the result part could be minimized to the figure and table placed as a supplementary file?)

L. 226:              Discussion

L. 242-248:      How can you be sure that high bruising scores are an effect of weight and not breed? Did you compare heavy and light pigs within GL pigs?                  

Comments on the Quality of English Language

English language revision by a native speaker or experienced scientific writer would make the paper easier to read.

Author Response

Dear Reviewer 1,

thank you very much for reviewing our manuscript and for giving these supporting comments. We have tried to consider all comments, improved the introduction, and presented the results and conclusion more clearly.

Additionally we tried to give more weight to the results and focused on a clear structure throughout the manuscript.

We hope that fits to your ideas.

With my best regards, on behalf of all authors

Maren Bernau

Reviewer 2 Report

Comments and Suggestions for Authors

This study assessed the prevalence and extent of foot lesions and claw Angle growth in piglets of three different genetic lines from the date of birth to the end of lactation. The results showed that bruising and dorsal horn lesions reached the highest level on day 8 and then declined; There are differences between genetic lines. Before weaning (day 28), the front foot was more affected by bruising than the back foot, but by day 71, the back foot had a higher percentage of bruising than the front foot. The proportion of dorsal Angle lesions in the external claw was higher. In addition, from birth to the end of seedling, there were significant differences in claw Angle length among genetic lines.

The topic of this paper is relatively novel, and the entry point is appropriate. however, as for the feeding supplementation information is lacking in the methods, and a confused design for the different farrowing experiences and ages of sows, further refinement and modification are needed in the experimental design. The details are listed as follows:

1.         The wording of foot lesions should be standardized throughout the text. For example, "piglet feet lesions" in the title and "Piglet´s foot lesions" in line 12. Similar ones in the text should be changed to a uniform writing style.

2.         In line 64, the feeding supplementation of piglets during the test period was not explained in the animal feeding management section. Since the daily gain of piglets was recorded in this experiment, the feed supplementation has an unavoidable effect on daily gain and should be noted in the paper.

3.         In lines 67-68, length and width data punctuation errors should be changed to the decimal point.

4.         There is no space between multiple P-values and symbols, as in line 142.

5.         According to animal ethics, the text should clearly state what happens to the animals after the experiment.

6.         In line 91, what is the basis for choosing to measure hooves on days 3 and 8 of the piglet's life? The basis should be given;

7.         In lines 167 and 172, the description of the two sentences was disagreement about whether there was a significant difference between male and female piglets.

8.         In line 314, as for the reasons for the high back angle damage, the effects of fighting and trampling in the litter should be considered in addition to the effects of floor conditions. The conjectures are not strong enough to support your point. For instance, if the specific heavier genetic pigs may show more trampling than others, how can you dismiss such effects? All the possibilities should be indicated.

9.         Figure 3 is vague, which is not up to the publication requirements, and should be changed.

10.     In Table 1, the farrowing times of GL-LW are different from those of the other two groups of sows. why do you choose the sows with different farrowing experiences (maybe with different ages) in the three groups? The farrowing experience affects the maternal performance of the sows which would impact the growth of piglets. The experiment should dismiss the effect of farrowing experience and the age of the sows. So, you must take care of the experience and age effect in consideration to discuss the differences among the three groups. Otherwise, the group with a different farrowing experience must be deleted to support your conclusion. Or, you could not conclude in the current form.

Author Response

Dear Reviewer 2,

thank you very much for reviewing our manuscript and for giving these supporting comments. We have tried to consider all comments, improved the introduction, and presented the results and conclusion more clearly.

Additionally we tried to give more weight to the results and focused on a clear structure throughout the manuscript, so that the methods part is less confusing. We keep this structure also for results and discussion section.

We hope that fits to your ideas.

With my best regards, on behalf of all authors

Maren Bernau

Round 2

Reviewer 1 Report

Comments and Suggestions for Authors

I still think the publication needs a major revision, both in the way the results are presented and analyzed.

The interesting part of this paper is the foot and claw lesions over time.

Consider leaving out weight, weight gain and claw horn length as outcome variables in your paper. These topics are not part of your initial aim, and as far as I can see, you do not relate weight or claw horn length to foot and claw lesions in a mathematical way. If you leave them in your paper, you need to explain why weight, weight gain and claw horn length of 74 pigs is new knowledge and important for research and pig production in general.

 I suggest you discuss at what level the foot and claw lesions you investigate are of clinical and pathological importance.

Please reconsider the statistical analyses: are all explanatory variables accounted for in the mathematical analyses for foot and claw lesions? I.e. mother sow, weight? You might need to consult an epidemiologist on the matter.

Title + manuscript in general: Since results from both suckling and nursery pigs are included, please replace ‘piglets’ with ‘pigs’, when referring to both age groups.

L. 23-24:           delete “and decreased afterwards.”

L. 36:                  Please rephrase: Foot lesions in piglets have been discussed for several years.

L. 37:                  Please provide a reference for foot lesions as a present problem.

L. 62-63:           “Additionally, weight gain and claw horn growth were evaluated in order to check the association to claw lesions” As far as I can see you do not test these associations statistically.

L. 66:                  2.1 “animals and housing….” Please start with a capital A. In general, please start all headings with a capital letter.

L. 77:                  Please replace “pregnant sow units” with “gestation units”

L. 95-97:           Did the piglets stay with their own mother in the total suckling period? Or could they be moved to a different sow? Please specify.

L. 109:                Do you by ‘alteration’ mean ‘bruises’ or ‘lesions’? Please rephrase.

L. 119:               similar as L: 109.

L. 144:               You use a Fisher’s exact test to test the association between bruises and genetic line, However, you report a general statistically significant different between genetic lines. Please also report which lines are different GL vs GL-LW?, GL vs Pi-PiDU? etc.

L 160                  Did you try out mother sow, as fixed effect in your model?

L. 163:               Please do the model again with birth weight as a covariate in your weight model, as other studies have shown statistically significant impact of birth weight on weight in pigs.

L. 166:               Results in general: Start by presenting demographics and descriptive statistics:

How many pigs were included in total, how many did you find again at the different time points, how many died and how many were unaccounted for (missing observations). 

You mention 74 pigs initially, but only 67 to 58 appear in figure3. What happened to the 6 pigs?

L. 168:               Please move the first sentence to the Materials and Methods chapter.

L. 169:               Please mention the exact association you are referring to (I guess palmar surface front and rear feet). Did you dichotomize your data in the statistical analyses (i.e in bruising yes/no) When you mention statistically significant associations, please mention the results (or refer to a table), and also write the p-values.

L. 171-172:      You write: “On day 0, bruising is only found in less than 90% of the animals”. Please write the exact number, and leave out the word ‘only’.

L.173-173:       “with focus on the palmar surface of especially the front claws (score 2: inner claw 13.43%, outer claw 10.45%)”. Do you mean that 13.4 and 10.5 % of pigs have a score 2 or less than 2? I do not understand this sentence.

In general: Does it make sense to use more than one decimal in reporting your results?

L. 187:               You refer to Table S1. Bruising score results: How are these calculated? I guess you add up the scores, but you need to explain how in chapter 2.2.1. OR are the numbers in the table percentages? What is 0,1,2,3,4?

                             Please also include numbers of observations in the table.

L. 217:               Again, please write the number og pigs included for analyses.

You show statistically significant differences of what (weight or weight gain?) With how much, and what is the p-value?

L.: 239:              “Claw development shows no clear pattern among genetic lines” I do not understand. Do you mean that you could not find any statistically significant differences between genetic lines?               

L. : 241               day 70 or 68? You use 68 in the table. Please be consistent.

L. : 370              “The present study found significant differences among the genetic lines until day 68 (±2) of life, with GL-LW having the shortest claw horn length (Table 5; at some days together with Pi-PiDu), but also having the lightest piglets (Table 3). Whereas, GL had the longest claw horn (Table 5, front and rear legs), combined with the heaviest piglets”

Did you test this statistically?

You state something different in L 241.

Comments on the Quality of English Language

Moderate revision of the English and scientific language is needed. 

Author Response

Dear Reviewer 1,

thank you very much for reviewing our manuscript again and for giving these supporting comments. We added a logistic stepwise regression analysis to try to explain the effects on bruising scores. This additional statistic illustrates the importance of the individual parameters and explains that, among other things, the claw horn length has an influence on the bruising process. We tried to consider all comments and hope that fits to your ideas.

With my best regards, on behalf of all authors

Maren Bernau

Reviewer 2 Report

Comments and Suggestions for Authors

All questions have been answered and clearly explained, and the revisions are sufficiently detailed and complete that I have no further questions.

Author Response

Dear Reviewer 2,

thank you so much for your supporting review. Due to comments of reviewer 1, we added a logistic stepwise regression analysis to try to explain the effects on bruising scores.

This additional statistic illustrates the importance of the individual parameters and explains that, among other things, the claw horn length has an influence on the bruising process.

We hope you agree and look forward to your feedback.

On behalf of all authors and with my best regards

Maren Bernau